# Management of Birch Spruce Mixed Stands with Consideration of Carbon Stock in Biomass and Harvested Wood Products

Jānis Vuguls, Stefānija Dubra *, Anete Garanča, Daiga Zute  and Āris Jansons

Latvian State Forest Research Institute Silava, Rigas Street 111, LV-2169 Salaspils, Latvia
* Correspondence: stefanijadubra@gmail.com

**Abstract:** Forests play an important role in climate change mitigation. Usage of harvested wood products (HWP) can extend the carbon cycle by retaining carbon as well as preventing new fossil emission via substitution. We compared carbon balance of different management strategies of birch spruce mixed stands over an eight-year period: unmanaged, representing a decision of prolonged rotation, and managed, representing a decision of final harvest of birch and retention of spruce for continuous forest cover and regeneration harvest. Management resulted in a higher contribution of mixed stands to climate change mitigation, if the carbon stock (CS) in biomass as well carbon balance (CB) of wood product is jointly considered in comparison to no management (prolonged rotation). Assortment structure plays an important role in CB of HWP, therefore a practice ensuring higher outcome of longer-lasting wood products are beneficial to climate change mitigation.

**Keywords:** carbon balance; carbon pools; climate smart management; harvested wood products; substitution effect



## 1. Introduction

Climate change has been a globally discussed topic for the last decades, especially in terms of mitigation of greenhouse gas (GHG) emissions. One of the European Union's (EU) goals in the aspect of climate change is to achieve a climate neutral situation (zero emissions of carbon dioxide ($CO_2$)) by 2055 by reducing GHG emissions and increasing C sequestration from the atmosphere [1]. The aim of Climate and Energy Framework is to reduce 40% of GHG emissions by 2030 for all sectors as a part of the Paris Agreement [2]. In addition, the European Commission (EC) published a proposal for incorporating greenhouse gas emissions and removals due to the Land Use, Land Use Change and Forestry (LULUCF) into its 2030 Climate and Energy Framework [3].

Climate smart forestry (CSF) is a targeted and long-term forest use strategy with a focus on mitigation of climate change, and an essential part of CSF is assimilation of $CO_2$. Forest carbon pools consist not only of living tree biomass (above- and belowground), but carbon can also be stored in harvested wood products (HWP) after felling [4,5]. While carbon pool of HWPs is recognized as marginal compared to existing boreal and temperate forest ecosystem [6,7], one of the CSF spectrum is the usage of HWP instead of emissions-intensive materials [4,8]. It helps to avoid from additional emissions caused by production of fossil-based materials [9]. In addition, the efficient use of HWP through optimal allocation of tree parts is important to avoid from losses in biomass and energy. For example, harvested trees can be primarily used as construction wood followed by recycling the post-consumer lumber waste. Another example is that the paper production cascade can be built on residues from wood chips or pulpwood, followed by waste paper recycling [10]. Furthermore, a study in Poland showed that forest waste biomass can be suitable raw material for briquette production thereby reducing the emission of pollutants into the atmosphere [11].

Various studies have shown that forest carbon stocks might be maintained and even enhanced by different type of forest management because wood is used in various assort-

ments based on wood quality, species and tree dimensions [8,9]. There have been different perspectives on carbon stock analysis to different forest management approaches and stand age. Furthermore, carbon balance of forest stands must be analyzed in the long term, because after felling the carbon balance will always be negative in a short-term period [12]. Sustainable forest management is a part of the CSF, and thinning has been proposed as a way to enhance aboveground carbon storage by simultaneously and positively affecting ground vegetation and favoring growth of productive timber logs [10,11]. Since it is well known that high-intensity thinning reduces organic carbon in the soil [13], there have been recommendations to promote continuous cover forestry (CCF) instead of natural regeneration to maintain optimal tree density [14,15]. CCF is known as close to nature forestry (a silvicultural system without a clearcut phase), and besides the European Union (EU) Biodiversity Strategy recommends to develop close-to-nature forest management practices [16]. In addition, carbon accumulation in trees is species specific, but on the other hand, a study in Canadian boreal forests has demonstrated that the quantity of stored carbon depends on specific species composition, representing complementary distribution of resources for nutrients, water and light [17]. Besides, in temperate forests carbon losses after harvesting are smaller in softwood–hardwood mixed stands (20%) than in monospecific hardwood stands (36%) [18].

There have been several studies about carbon stock in mixed and single species stands [16–19] as well as studies about the influence of thinning on carbon pools [5,9–11] and carbon balance in HWP [5,9]. To achieve the targets of Paris agreement and climate neutral situation, country-specific studies about carbon balance and use of HWP are urgent, because the use of wood products is an important strategy to mitigate climate change. Besides, reduction plans to mitigate GHG emissions differ between EU countries since each country has specific natural and economic resources. For instance, forests are one the most important natural resources in Latvia which constitutes a major part of the total carbon pool. Thus, the aim of our study is to evaluate the short-term (8 years) effect of forest management decisions (type and time of final harvest) on carbon stock in biomass and harvested wood products in birch spruce mixed stands.

## 2. Materials and Methods

### 2.1. Study Area

The study was conducted in hemiboreal forests in Latvia. The climate in this region is maritime. According to data of the Latvian Environment, Geology and Meteorology Centre, the mean annual air temperature is 7.06 °C and the annual precipitation is 683 mm. Based on the National Forest Inventory, the dominant species are birch (*Betula pendula* Roth. and *Betula pubescens* Ehrh.), Scots pine (*Pinus sylvestris* L.) and Norway spruce (*Picea abies* (L.) Kast.). Prevailing mixture of species include birch and Norway spruce.

### 2.2. Sampling and Measurements

Sample plots with an area of 0.2 ha were placed in randomly selected birch spruce mixed stands in forest types suitable for growth of both tree species—21 stands on fertile fresh mineral soil (*Hylocomniosa*, *Oxalidosa*, *Aegopodiosa* forest types), and 8 stands on drained mineral soil (*Myrtillosa* mel., *Mercurialiosa* mel.) [20]. Spruce as shade tolerant species was suppressed, its mean diameter at breast height (DBH) being on average 0.7 of that of birch.

After initial inventory, each sample plot was divided in two equal parts (Figure S1): unmanaged, representing a decision of prolonged rotation, and managed, representing a decision of final harvest of birch and retention of spruce for continuous forest cover (CCF). To represent the decision of regeneration cut (clearcut), we used measured tree data from the sample plots established in our study and regeneration data (parameters of young stands) from National Forest Inventory sample plots established at the same forest types.

Initial inventory was carried out before the treatment (harvesting of birch) in 2012 and repeated inventory after 8 years. In both inventories height and DBH for all trees, snags and

standing deadwood with diameter above 6.0 cm was measured. For all lying deadwood with diameter > 6.0 cm and length > 1.0 m we measured total length and diameter of the stem at both ends. We enumerated all lying and standing deadwood as well as snags by defining species if possible (if identification of species was not possible, we classified as at least deciduous or conifer) and decay stages using the National Forest Inventory (NFI) decay classification system:

0—Decay class represents raw wood. Trees that recently died, have not dried up.

1—Decay class represents solid dead wood. The stem has a hard-exterior surface, the volume of the stem consists at least 90% of solid wood.

2—Lightly decayed wood. Wood volume consists of 10–25% of softwood, solid remaining wood.

3—Decayed dead wood. The volume of stem consists of 26–75% of softwood/very soft wood.

4—Very decayed dead wood. The volume of stem consists of 76–100% of softwood/ very soft wood [21].

At the initial inventory increment cores were taken at stump height for ten dominant birch trees per stand and used to determine stand age. The mean stand age was 72 years (ranging from 51 to 89). For purposes of further analysis, we divided stands in two age groups: younger than 70 years (premature) and older than 70 years (mature).

### 2.3. Calculations

The tree biomass (above- and below-ground) carbon stock (CS) was estimated from the DBH and tree height for individual trees based on the local biomass equation [22]. The living tree biomass carbon stock was calculated using the living tree biomass values multiplied by the carbon content of 50% [23,24].

Volume of snags and lying deadwood were calculated according to the formula for a cylinder. The deadwood carbon stock was calculated based on deadwood volume estimations, decay class-specific density, and carbon content for the main tree species in hemiboreal forests [25].

To evaluate carbon balance (CB) the difference between carbon inputs and carbon outputs was of living biomass, dead organic matter and wood products were determined. CB of HWP from different birch and spruce assortments was calculated according to Pukkala [5] study. CB in HWP depends on several components: end use of product, decomposition rate, harvesting and manufacturing releases and substitution effects. Carbon stock in pool is calculated based on method from Pukkala studies [5,12] using decomposition rates from Row and Phelbs, 1990 published in Karjalainen et al., 1994 [4]. Positive balance means that forestry is carbon sink, but negative CB represents source of carbon. The unit of CB is Mg C ha$^{-1}$ a$^{-1}$ (per annum).

Assortment structure—percentage of saw logs, pulpwood, firewood (fuel feedstock made of stems)—was based on the dimensions of trees and obtained using methodology by Ozolins [26]. Substitution effect refers to the reduction in C releases from fossil fuels due to the use of forest biomass and wood products [27,28]. It differs between the wood product categories [4,5,12]. The used percentages of wood product categories from spruce and birch pulpwood are taken from the Pukkala [5] since in Latvia we do not have chemical processing and most of pulpwood is exported to Scandinavian countries, including Finland. Saw logs and biofuel are used in domestic processing, thus the percentages stated in Table 1 refers to situation in Latvia.

**Table 1.** Percentages of end product subgroups for saw log and pulpwood; *—end product percentages from Pukkala [12].

| Assortments | Sawn Wood | Mechanical Mass | Chemical Mass | Biofuel |
|---|---|---|---|---|
| Spruce saw log | 43 | 0 | 0 | 57 |
| Spruce pulpwood * | 0 | 76 | 8 | 16 |
| Birch saw log | 43 | 0 | 0 | 57 |
| Birch pulpwood * | 0 | 0 | 46 | 54 |

For data analysis used linear mixed-effects model analysis in the program R under version 4.0.3, library lme4 [29]. The predictor variables (tree species and stand level) were treated as fixed effects. To assess the differences in carbon stock between managed and unmanaged stands as well as between stand age we used the analysis of variance (ANOVA) as implemented in program R. We performed Tukey's Honestly Significant Difference (Tukey's HSD) post-hoc test to assess significant difference between analyzed groups.

## 3. Results

Removal of birch had results in growth release: eight years after the CCF treatment in managed stands the mean diameter at breast height, the mean tree height and the mean basal area for spruce trees were higher than in unmanaged stands (Table 2), but the mean volume of spruce trees between managed and unmanaged stands was similar. Birch trees as a helophyte species reached the highest stand parameters in unmanaged stands.

**Table 2.** Characteristics of mixed (unmanaged) and monospecific (managed) stands, where for unmanaged stands prevailing tree species are *Betula pendula* Roth. and *Picea abies* (L.) Karst., for managed, *Picea abies* (L.) Karst. DBH—diameter at breast height. CI—95% confidence interval.

| Parameters $\pm$ CI | Unmanaged Stands | | Managed Stands |
| | Spruce | Birch | Spruce |
|---|---|---|---|
| Mean tree DBH, cm | $15.23 \pm 0.30$ | $22.74 \pm 0.44$ | $18.14 \pm 1.45$ |
| Mean tree height, m | $14.28 \pm 0.23$ | $24.83 \pm 0.31$ | $15.58 \pm 1.20$ |
| Mean basal area, m$^2$ ha$^{-1}$ | $15.31 \pm 1.80$ | $21.61 \pm 2.09$ | $17.55 \pm 1.88$ |
| Mean stand volume, m$^3$ ha$^{-1}$ | $139.43 \pm 18.52$ | $269.18 \pm 30.58$ | $139.68 \pm 21.99$ |
| Number of sample plots | 29 | 29 | 29 |

### 3.1. Carbon Stock between Different Management Practices

In the first scenario we compared carbon stock between two different age groups of managed (CCF) and unmanaged stands: <70 (premature) and >70 years (mature). The carbon stock in managed sample plots (C in living tree biomass + HWP) was significantly higher ($p < 0.05$) than carbon stock in unmanaged sample plots (only C in living tree biomass). In the premature managed stands the mean stock of living tree biomass was $6.7 \pm 1.2$ Mg C ha$^{-1}$, but in unmanaged $15.1 \pm 2.2$ Mg C ha$^{-1}$. In mature managed stands the mean stock of living tree biomass was $5.8 \pm 1.0$ Mg C ha$^{-1}$, but in unmanaged $15.9 \pm 1.6$ Mg C ha$^{-1}$ (Figure 1a). In premature managed sample plots the CB of HWP was $11.5 \pm 1.9$ Mg C ha$^{-1}$, but in mature stands CB was $15.3 \pm 3.0$ Mg C ha$^{-1}$. In the second and third scenario we compared the carbon stock between two different age groups of managed (clearcut, regenerated with Norway spruce or silver birch) and unmanaged stands. Carbon stock of both Norway spruce and silver birch young stands + carbon stock of HWP was significantly higher ($p < 0.05$) than living tree biomass in unmanaged sample plots (Figure 1b,c). The mean CB of HWP was $18.7 \pm 0.8$ Mg C ha$^{-1}$. According to the National Forest Inventory, the mean carbon stock of young Norway spruce stand is $8.9 \pm 2.0$ Mg C ha$^{-1}$, but for silver birch $7.0 \pm 1.3$ Mg C ha$^{-1}$, but carbon stock in HWP was higher in the silver birch ($13.7 \pm 2.0$ Mg C ha$^{-1}$) than the Norway spruce ($5.0 \pm 0.8$ Mg C ha$^{-1}$).

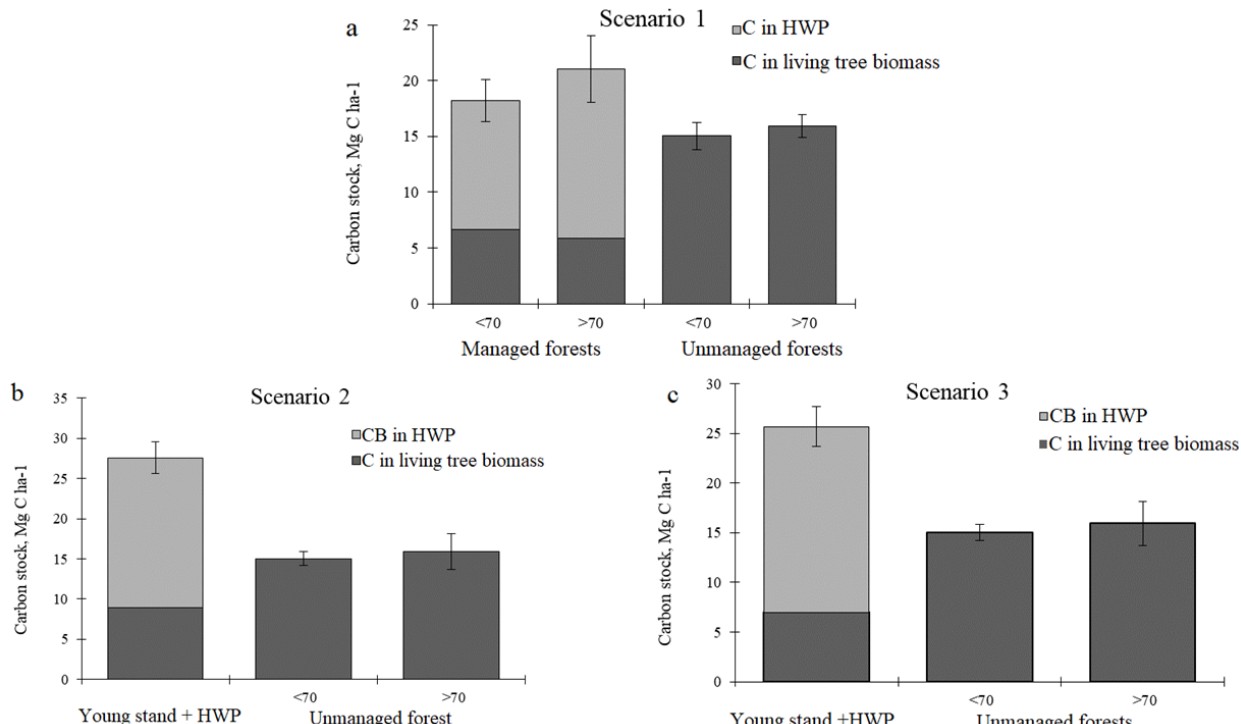

**Figure 1.** Comparison of carbon stock between managed and unmanaged stands. In managed stands carbon stock represents carbon in living tree biomass plus carbon balance in harvested wood products eight seasons after treatment. (**a**)—comparison of carbon stock between stands treated with continuous cover forestry and unmanaged stands; (**b**)—comparison of carbon stock between eight-year-old Norway spruce stand (carbon of eight-year-old stand + carbon balance of HWP) and unmanaged (mixed-species) stands of two age groups; (**c**)—comparison of carbon stock between eight-year-old silver birch stand (carbon of eight-year-old stand + carbon balance of HWP) and unmanaged (mixed-species) stands of two age groups. CB—carbon balance; HWP—harvested wood products, <70—premature stands (younger than 70 years); >70—mature stands (older than 70 years).

### 3.2. CB of HWP

The CB of HWP was calculated using four components: carbon mass in new products, decomposition, harvesting and manufacturing releases and substitution effects. The volume of assortment outcome varied among species. For silver birch the volume of saw log assortment outcome was 213 m$^3$, higher quality saw logs 152 m$^3$, pulpwood 118 m$^3$, packing case timber 113 m$^3$ and firewood 12 m$^3$. For Norway spruce the volume of pulpwood assortment outcome was 108 m$^3$, packing case timber 97 m$^3$, saw logs 81 m$^3$, higher quality saw logs 24 m$^3$ and firewood 8 m$^3$. Despite the assortments, for silver birch the highest CB was in high-quality saw logs, but the lowest for firewood (Figure 2b). The CB of silver birch HWP was $3.9 \pm 1.0$ Mg C ha$^{-1}$ a$^{-1}$ in higher-quality saw logs, $5.4 \pm 0.9$ Mg C ha$^{-1}$ a$^{-1}$ in saw logs, $2.9 \pm 0.3$ Mg C ha$^{-1}$ a$^{-1}$ in packing case timber, $1.3 \pm 0.2$ Mg C ha$^{-1}$ a$^{-1}$ in pulpwood and $0.3 \pm 0.1$ Mg C ha$^{-1}$ a$^{-1}$ in firewood. The CB of Norway spruce HWP was $0.5 \pm 0.2$ Mg C ha$^{-1}$ a$^{-1}$ in higher-quality saw logs, $2.1 \pm 0.5$ Mg C ha$^{-1}$ a$^{-1}$ in saw logs, $1.8 \pm 0.3$ Mg C ha$^{-1}$ a$^{-1}$ in packing case timber, $0.6 \pm 0.1$ Mg C ha$^{-1}$ a$^{-1}$ in pulpwood and $0.1 \pm 0.01$ Mg C ha$^{-1}$ a$^{-1}$ in firewood (Figure 2a).

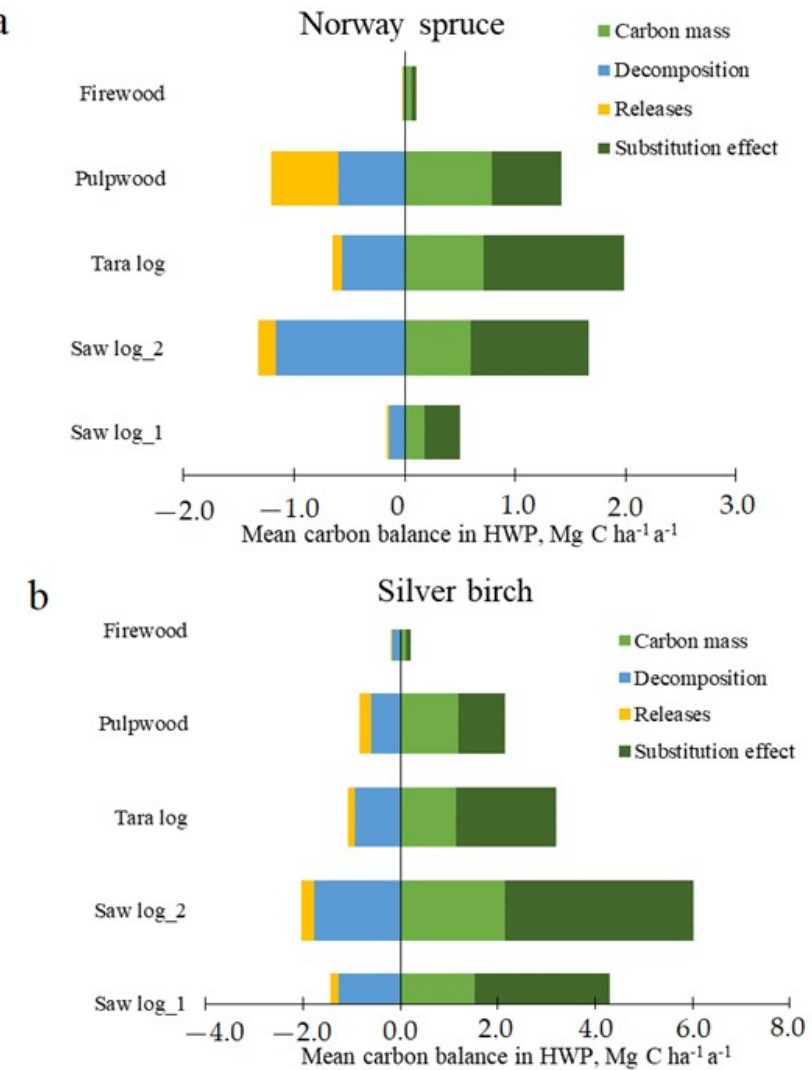

**Figure 2.** Carbon balance of harvested wood products. (**a**)—Carbon balance of harvested wood products of Norway spruce, (**b**)—Carbon balance of harvested wood products of silver birch. HWP—harvested wood products, HQ—high quality saw logs.

## 4. Discussion

In forest-based bioeconomy it is essential to promote climate change mitigation actions that provide carbon storage in living tree biomass and HWP [25,30,31]. Interactions between sustainable development and financial benefits is a key to achieve climate neutral scenario [25]. Previous studies have shown that in the long-term old-growth forests cannot be robust carbon sinks, although net carbon quantity can be positive because in managed stands carbon is stored in HWP either in the next generations [9,25]. Our results suggest that based on CB of HWP managed stands with clear cuts and continuous cover forestry provide higher long-term carbon stock compared with unmanaged stands (Figure 1). In sustainable forest management, it is necessary to carry out thinning. It opens up space for the remaining growing trees, stand can reach productive timber logs, it positively affects vegetation and increases species abundance and diversity in most forest site types [10,11]. On the other hand, high-intensity thinning can significantly reduce the soil organic carbon storage [13]. Our results show that 8 years after forest management treatments CS is higher in young stands than in stands treated with CCF. Accordingly, the mean carbon stock in Norway spruce young stand was $8.9 \pm 2.0$ Mg C ha$^{-1}$, but CS in remained Norway spruce living tree biomass was $6.2 \pm 0.7$ Mg C ha$^{-1}$. CS in living biomass + HWP were significantly ($p < 0.05$) higher in managed stands compared to unmanaged in all management practices

represented in our study. Previous studies suggest that uneven age management and high intensity thinning provides better total C assimilation compared to low-intensity thinning and even-aged forestry [5]. It is explained by usage of HWP—harvested wood mainly was used in long-lived products and construction purposes. In addition, the collection of harvest residues is easier in clear felling sites [32,33]. Besides, the HWP carbon balance includes changes in carbon pools in wood-based products, with regard to emissions from harvesting, manufacturing and substitution effects when replacing non-wood products produced from fossil fuels with wood-based production [25,31].

One of the forest management practices included in study was clear cuts regenerated with single species stands. Our data showed that young stands have higher carbon stock compared to mature and premature stands (Figure 1) and it is species specific, indicating that clear cut regenerated with Norway spruce provides higher carbon assimilation than with silver birch. Based on the wood structure, softwoods consist of more lignin compared to hardwoods, which explains higher carbon content in Norway spruce [23,24]. But our results showed that in case of carbon stock in HWP, silver birch had higher carbon stock compared to Norway spruce. It means that the most crucial component of carbon stock analysis in is the substitution effect and importance of reuse and recycle strategies. Each assortment category has its own substitution effect and decomposition rates. The substitution effects determine the impact of wood-based product use instead of non-wood products from fossil fuels and it is one of the most important indicators of climate change mitigation and GHG emission balance [31,34]. It alludes substitution rate of how much 1 mg of carbon in HWP reduces emissions from non-wood products [32]. Substitution effects in HWP varies between assortments. For instance, the substitution effect for sawn wood is higher compared to pulpwood or biofuel, because sawn wood products provide long term accumulation, respectively [35].

In previous studies the concept of carbon pools in HWP has been considered, proving that HWP is a suitable method for carbon assimilation and carbon stock. On the other hand, usage of HWP product cascade does not always lead to significant carbon stock. Previous studies have shown that quality of wood product assortments can positively influence carbon pool in HWP [22,31]. We compared mean carbon stock between five different end wood product categories used in Latvia and our data suggest that wood product assortment category plays an important role in the mean carbon balance. Furthermore, the potential substitution benefits of HWP depend on the quality of wood and wood usage efficiency during its lifecycle. For instance, our data show that the lowest substitution effect was determined for firewood, but the highest for saw logs and packing case timber. It is because the obtained emission reductions per unit of biomass for saw logs and packing case timber can be used for both material and energy substitution, and in the best scenario wood material can be recycled during its lifetime and only at the end used for energy. Firewood is used only for energy and obviously has lower unit benefits in emissions reduction. Besides, previous studies suggest that usage of HWP instead of emission-intensive materials helps to avoid additional emissions in atmosphere [9] and efficient use of HWP is important to avoid from losses in biomass and energy [10].

Previous studies mostly focus on single-species even-aged stands in terms of carbon sequestration and substitution effects, but our data represent changes in carbon stock between uneven-aged stands. Forest management practices and wood product assortment categories vary among different countries; therefore, it is not possible to consider one specific forest management approach to achieve positive CB. In the future country-specific studies have to be completed, including different type of forest management (even and uneven age) and wood-product-use strategies to understand carbon inputs and outputs in EU level.

## 5. Conclusions

In this study we compared CS between stands treated by different management styles. Management, considering the CS in biomass and CB of HWP, resulted in significantly higher contribution to climate change mitigation in comparison to no management. HWP has crucial role in this outcome. Comparison of CB between the most common wood product assortments in Latvia 8 years after felling: packing case timber, saw logs, pulpwood and firewood demonstrates the importance of striving towards the high-value wood products to further increase contribution of the forest sector to climate change mitigation.

**Supplementary Materials:** The following supporting information can be downloaded at: https://www.mdpi.com/article/10.3390/f14010057/s1, as Figure S1. Scheme of sample plots.

**Author Contributions:** Conceptualization, J.V. and D.Z.; methodology J.V.; data curation, A.G. and J.V.; writing—original draft preparation, J.V. and S.D.; writing—review and editing, Ā.J. and D.Z.; project administration, Ā.J.; funding acquisition, Ā.J. All authors have read and agreed to the published version of the manuscript.

**Funding:** This research was funded by ERDF project Development of a decision support tool integrating information from old-growth semi-natural forest for more comprehensive estimates of carbon balance (No 1.1.1.1/19/A/130).

**Conflicts of Interest:** The authors declare no conflict of interest.

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
