# Peer review of "Management of Birch Spruce Mixed Stands with Consideration of Carbon Stock in Biomass and Harvested Wood Products"

_forests, doi:10.3390/f14010057_

Round 1

Reviewer 1 Report

I find this manuscript interesting since it covers the effects of different management regimes and the carbon losses due to harvesting and decompostion. However, it is not clear how the carbon losses have been determined due to harvesting when the net primary productivity needs to be quantified as a surrogate measure for carbon loss. Besides, the living biomass which had been measured as a result of growth and metabolism can be considered as carbon losses upon harvesting. Moreover, these losses need to be accounted for and expressed over the eight years since harvesting. It is also not clear in the manuscript if the comparison between the living biomass of standing trees and those harvested come from the same stand depending on the management regime considered. I recommend accepting the manuscript after the comments and recommendations have been considered.   

Reviewer 2 Report

After careful reading, I consider that this manuscript's structure, logical flow, literature review, and statistics are not up to the standards. I found high similarities with published literature on the internet, which are similar to this study. Authors made frequent mistakes throughout the MS.

I noticed that a separate conclusion section is totally missing, and seems the authors have included it in the discussion. I suggest writing separate conclusion section for clarity. 

The authors would do well to refer to other peer-reviewed publications for guidelines on what is most appropriate in tables, results, and figures and what is better placed in an appendix.

Specific suggestions are included in the ZIP file 

Reviewer 3 Report

The article is extensive, but it does not cover the use of all alternative energies. For example, other sources of alternative energy should be mentioned. Please refer to this article and discuss it in your manuscript, for example, in terms of energy expenditure https://doi.org/10.3390/en14113270.

There is no in-depth statistical analysis. Statistical analysis was not sufficiently discussed. The entire manuscript looks like an excerpt from a report from some major work. Without statistics and research, the article does not contribute to any knowledge development. An expansion of literature, conclusions, and discussions is required.

Round 2

Reviewer 2 Report

Authors considered reviewers suggestions and did extensive edits than initial submission. The subject scope is within the scope of the journal. The title and abstract accurately describe the contents. The abstract include all the main ideas presented in the article. Structure of the manuscript seems correct. The results are discussed with other presented in literature (last 3 years).